# The Regulatory Mechanism of Cold Plasma in Relation to Cell Activity and Its Application in Biomedical and Animal Husbandry Practices

**DOI:** 10.3390/ijms24087160

**Published:** 2023-04-12

**Authors:** Yijiao Wu, Shiyu Yu, Xiyin Zhang, Xianzhong Wang, Jiaojiao Zhang

**Affiliations:** Chongqing Key Laboratory of Forage and Herbivore, College of Veterinary Medicine, Southwest University, Chongqing 400715, China

**Keywords:** cold plasma, cell activity, wound healing, cancer therapy, animal husbandry application

## Abstract

As an innovative technology in biological applications, cold plasma is widely used in oral treatment, tissue regeneration, wound healing, and cancer therapy, etc., because of the adjustable composition and temperature which allow the plasma to react with bio-objects safely. Reactive oxygen species (ROS) produced by cold plasma regulate cell activity in an intensity- and time-dependent manner. A low level of ROS produced by cold plasma treatment within the appropriate intensities and times promotes proliferation of skin-related cells and increases angiogenesis, which aid in the acceleration of the wound healing process, while a high level of ROS produced by cold plasma treatment performed at a high intensity or over a long period of time inhibits the proliferation of endothelial cells, keratinocytes, fibroblasts, and cancer cells. Moreover, cold plasma can regulate stem cell proliferation by changing niche interface and producing nitric oxide directly. However, the molecular mechanism of cold plasma regulating cell activity and its potential application in the field of animal husbandry remain unclear in the literature. Therefore, this paper reviews the effects and possible regulatory mechanisms of cold plasma on the activities of endothelial cells, keratinocytes, fibroblasts, stem cells, and cancer cells to provide a theoretical basis for the application of cold plasma to skin-wound healing and cancer therapy. In addition, cold plasma exposure at a high intensity or an extended time shows excellent performances in killing various microorganisms existing in the environment or on the surface of animal food, and preparing inactivated vaccines, while cold plasma treatment within the appropriate conditions improves chicken growth and reproductive capacity. This paper introduces the potential applications of cold plasma treatment in relation to animal-breeding environments, animal health, their growth and reproduction, and animal food processing and preservation, which are all beneficial to the practice of animal husbandry and guarantee good animal food safety results.

## 1. Introduction

Plasma, as the fourth form of matter, is composed of a high number of ionized active particles, such as charged ions, free electrons, reactive atoms and molecules, free radicals, and unionized neutral particles. Plasma emits electromagnetic radiation including infrared, visible and ultraviolet photons, and radiates transient electric fields [1]. Cold plasma presents a low-temperature state (<40 °C) because it is characterized by the fact that the temperature values of ions and neutral particles are much lower than those of electrons [2]. Synergistic action with transient electric fields, charged species, or ultraviolet photons may evoke a unique feature for some of the plasma-based biomedical applications [3]. As a result of the advancing studies being conducted on plasma technology, cold plasma is being widely adopted in biomedical fields, such as oral treatment [4], tissue regeneration [5], skin-wound healing [6,7], and cancer therapy [8,9,10]. The type of energy input, input voltage or discharge power, gas component, gas pressure, and radiation type of electric field determine the exact composition and property of cold plasma [11]. The detailed analysis of gas and liquid phase chemistries, the generation of energetic photons, and the parameters of electric fields are required for the mode of plasma action and the development of plasma sources for specific applications.

When tissue is damaged, local immune cells and fibroblasts secrete growth factors, such as fibroblast growth factor (FGF) and epidermal growth factor (EGF), to increase the proliferation of keratinocytes and endothelial cells and to promote the regeneration of tissues [12]. Reactive oxygen species (ROS) are also an important factor that affects cell activity. A low level of ROS promotes cell proliferation by affecting Cyclin D1, while excessive ROS can induce cell cycle arrest and apoptosis by damaging the function of DNA and mitochondria in the body [13]. External factors, such as ionizing radiation and pulsed electromagnetic fields, can regulate cell activity by affecting the release of growth factors and the level of intracellular ROS. Cold plasma-producing reactive oxygen and nitrogen species (RONS), nitric oxide (NO), and electrons may cross a cell membrane barrier to reach the cell interior through a membrane channel, a passive diffusion process, or pores generated by lipid oxidation [14]. Although it is unclear how the RONS are delivered into the real biological targets and the exact interaction with various components of a tissue when considering the lifetime, diffusion rate, and major physical barriers to traverse [14], the diffusion and delivery of plasma-generated RONS or stimulation of intracellular RONS generating mechanisms as a result of cold plasma treatment have been suggested to regulate cell activity in both intensity- and time-dependent manners [15]. ROS produced by cold plasma treatment at the appropriate intensity and time induce the release of FGF-2 and promote endothelial cell proliferation, while a high amount of ROS produced by cold plasma treatment at a high intensity or over a long period of time inhibits the proliferation of endothelial cells [15], keratinocytes [16], and fibroblasts [17]. In addition, cold plasma treatment administered at a low intensity and over a short period of time had no significant effect on the activity of normal cells, while it induced cancer cell growth arrest and apoptosis [9]. Therefore, the different effects of cold plasma treatment on cell activity are related to the level of ROS produced under different treatment conditions.

In addition to its application in biomedical fields, cold plasma is also mainly used, at present, in the processes of sterilization [18], biological purification [19], vaccine production [20], and food processing [21]. Our previous studies showed that appropriate cold plasma treatment promoted the development of chicken embryos [22], the growth rate of chickens and the reproductive performance of roosters [23,24,25,26], and affected the proliferation of Sertoli cells in chicks [27]. These effects of cold plasma treatment may be related to its radiation type (including electric field and ultraviolet radiation) and its components (ROS, reactive nitrogen species (RNS), NO, and electrons, etc.) [28]. However, the application and potential mechanisms of cold plasma treatment in the field of animal husbandry remain unclear. Therefore, this study reviews the effects and the possible regulatory mechanisms of cold plasma treatment on the activities of endothelial cells, keratinocytes, fibroblasts, stem cells, and cancer cells to provide a theoretical basis for the application of cold plasma treatment to skin-wound healing and cancer therapy in human beings and animals. In addition, this paper also introduces the potential applications of cold plasma treatment in relation to animal-breeding environments, animal health, their growth and reproduction, as well as animal food processing and preservation, which are all beneficial to the practice of animal husbandry and ensure the safety of animal food.

## 2. Application of Cold Plasma Treatment in Biomedical Fields

In the literature, two types of cold plasma treatment are generally considered: plasma jet and dielectric barrier discharge (DBD) [29]. In a plasma jet, the electrode (pin, ring, or plate type) is located in a capillary or tube inside a pen-like device, the plasma is generated inside the device, and the effluent is blown out along the gas flow through the tube and can directly contact with target. In a DBD, the plasma is generated in the surface around a high voltage or in the gap between an isolated high voltage electrode and target. These two types of cold plasma are generated inside small discharge gaps, consisting of transient micro discharges or filaments with non-uniform and constricted property [30]. CINOGY GmbH Plasma Derm DBD, AdtecSteriPlas^®^ DBD [31] and MED NTPJ [29] are IIa medical devices that have been certified by CE in Germany. Cold plasma technology has been extensively used in the fields of sterilization [18,32], blood coagulation [33], inactivated vaccine preparation [20], biofunctionalized materials [34,35,36], and oral treatment [4]. In addition, low-intensity cold plasma treatment administered over a short period of time can successfully promote cell proliferation activity; therefore, it is used to promote angiogenesis, tissue vascularization, wound healing, skin repair, and tissue regeneration. High-intensity cold plasma treatment administered over a long period of time can inhibit cell activity and promote apoptosis; therefore, it has a positive effect on the treatment of cancers and reducing scar formation (Table 1). Therefore, it can be observed from the results that the effect of cold plasma treatment on cell activity presents a good application prospect in biomedical fields.

## 3. Effect of Cold Plasma Treatment on Skin-Related Cell Activity

The skin is the first barrier in the body to resist the invasion of external pathogens. Severe skin injury is often accompanied by wound infection, resulting in a slow wound healing process. At present, there are more than 38 million patients suffering from chronic wounds in the world [45]. The promotion of good skin-wound healing and the best methods to achieve this result remains a crucial issue in the research being conducted to date in the field of dermatology. In recent years, 3D bioprinters and nanocomposites have shown good application prospects in wound healing and tissue regeneration [46,47,48]. Besides, cold plasma technology has also been observed to effectively promote the activities of skin repair and wound healing [6,7]. Wound healing is an accurate and complex process that includes the hemostatic/inflammatory, proliferation, and remodeling stages. In the proliferation stage, new granulation tissues and blood vessels formed by the proliferation of endothelial cells, keratinocytes, and fibroblasts are the key to the wound healing process [49]; therefore, it is very important to regulate the proliferative ability of those cells during this stage.

### 3.1. Effect of Cold Plasma on Endothelial Cell Activity

Neovascularization is a process that is essential for successful wound healing. Endothelial cells are the key components located in the inner wall of blood vessels. A variety of growth factors, such as FGF, angiopoietin (Ang), vascular endothelial growth factor (VEGF), and transforming growth factor (TGF), have a direct or indirect regulatory effect on the proliferation of endothelial cells [50]. FGF-2 can regulate the proliferation of endothelial cells through the rat sarcoma (RAS)/mitogen-activated protein kinase (MAPK) pathway after binding to the fibroblast growth factor receptor (FGFR) [51]. VEGF can promote angiogenesis through the RAS-MAPK-extracellular regulated protein kinase (ERK) pathway after binding to the vascular endothelial growth factor receptor (VEGFR) [52]. Compared with a 3D bioprinter which is conducive to endothelial cell adhesion and generation and improvements of blood vessels and endothelial barrier [46,47], as well as nanocomposite scaffolds which enhance biocompatibility and provides a suitable environment for cell growth by adsorbing growth factors [48], the effect of cold plasma on endothelial cell activity is mainly mediated by different levels of ROS which are produced under different treatment conditions. In their study, Kalghatgi et al. [15] determined that low-intensity cold plasma treatment (DBD, discharge power at 4 J cm^−2^) conducted over a short period of time (30 s) could promote endothelial cell proliferation through ROS-mediated FGF-2 release, while treatment administered at a high intensity (discharge power more than 8 J cm^−2^) or over a long period of time (more than 60 s) resulted in endothelial cell death (Figure 1). Moreover, in a study conducted by Arjunan et al. [53], they observed that cold plasma treatment generated ROS in an intensity-dependent manner (DBD, discharge power increased from 4.2 to 8.4 J cm^–2^, and intracellular ROS increased from 14 to 22 percent compared with untreated cells), and further promoted endothelial cell proliferation, migration, and angiogenesis by inducing the release of FGF-2. In addition, cold plasma treatment can induce the secretion of FGF-2, Ang-2, and VEGF, and increase the expressions of FGFR and VEGFR, which further promotes angiogenesis during the wound healing process [31] (Figure 2).

### 3.2. Effect of Cold Plasma on the Cell Activity of Keratinocytes

Keratinocytes are the main components of the epidermis, and their proliferation is the main factor affecting tissue repair and regeneration processes [54]. In their study, Wende et al. [16] observed that cold plasma (plasma jet, discharge power increased from 2 to 6 kV) treatment time of less than 10 s had no significant effect on the activity of keratinocytes; however, when the treatment time was longer than 30 s, the number and activity of keratinocytes decreased with the increase in time (Figure 1). The main reason for this is that excessive ROS produced by cold plasma treatment at a high intensity or over a long period of time not only terminated the proliferation of keratinocytes in the G2/M phase, but also cause the oxidative damage of DNA and lead to cell apoptosis. H_2_O_2_ can induce the synthesis of interleukin-8 (IL-8) in human bronchial epithelial cells by activating the MAPK/ERK pathway and further affect cell activity [55]. The MAPK/ERK signaling pathway can promote cell proliferation and inhibit apoptosis by affecting the expression of Cyclin D1, apoptosis-related proteins, and other effector molecules in the G1/S phase [56]. Cold plasma can trigger ERK phosphorylation through the production of ROS and RNS, thus activating Cyclin D1 and cyclin-dependent kinases, and further promoting the proliferation of keratinocytes and accelerating the skin-wound healing process [57] (Figure 2). In addition, intracellular K^+^ leakage can activate ERK [58]. In their study, Hotta et al. [59] observed that cold plasma-treated medium (PN-120 TPG, treatment time of 2 min) activated ERK by stimulating intracellular K^+^ leakage, thus inducing the expression of IL-8 mRNA and promoting the proliferation of human keratinocytes.

### 3.3. Effects of Cold Plasma on the Cell Activity of Fibroblasts

In the process of wound healing, proliferated fibroblasts form a temporary extracellular matrix by producing and depositing collagen and fibronectin, which support and provide the nutrition for tissue proliferation during the wound healing process. However, the continuous proliferation of fibroblasts and excessive deposition of the extracellular matrix can cause hypertrophic scars. H_2_O_2_ and NO generated by cold plasma treatment (DBD, voltage at 14 kV and treatment time of 300 s) can cause the acidification (pH 6.7) of the PBS buffer, thus inhibiting the proliferation of human foreskin fibroblasts, which indicates that cold plasma may have a good therapeutic effect on hyperplastic skin diseases [17]. Fibroblasts can synthesize and secrete growth factors, such as EGF, TGF, and FGF, which can directly stimulate the proliferation of fibroblasts [60]. When cells are exposed to growth factors, the residues of the inhibitory subunit of nuclear factor kappa B alpha (IκBα) subunit on the p50/p65/IκBα trimer are phosphorylated, and the phosphorylated IκBα is ubiquitinated and degraded by the proteasome, which phosphorylates p65 and then induces the activation of nuclear factor kappa B (NF-κB) [61]. As a DNA-binding protein, NF-κB can promote the transition of cells from the G1 phase to the S phase by increasing the expression of Cyclin D1. Numerous studies have shown that mouse fibroblasts treated with cold plasma can induce ROS production and phosphorylate p65 by inhibiting the expression of IκB, activating the NF-κB signaling pathway, and increasing the DNA synthesis that occurs in the S phase, thus promoting fibroblast proliferation [37] (Figure 2). Cold plasma regulates the proliferation of fibroblasts in a time-dependent manner. Cold plasma treatment administered for an appropriate amount of time (DBD, 10 to 40 s) and intensity (discharge power at 1 W) can improve the proliferation of fibroblasts in wound tissues by inactivating the bacteria located around the wound, and ultimately shorten wound healing time; plasma treatment administered for a long period of time (more than 50 s) leads to cell apoptosis and necrosis, which inhibits wound healing activity; this may be related to the levels of ROS and RNS produced by the cold plasma [62]. In addition, plasma polymerization of allylamine on polypropylene membrane or plasma copolymerization of sulfobetaine methacrylate and acrylic acid in solutions were beneficial to cell adhesion and growth of mouse fibroblast cells, the mechanism might be related to OH radicals and other reactive species generations during the polymerization processes [34,35,36].

Briefly, cold plasma-inducing changes in the liquid environment of cells and reactive species generated in or transferred into cells, and plasma polymerization-creating biofunctionalized materials for culturing cells are identified as the main basic mechanisms of biological plasma action. Low-intensity cold plasma treatment administered over a short period of time is beneficial to the survival and proliferation of skin-related cells; however, high-intensity cold plasma treatment administered over a longer period of time can lead to cell proliferation inhibition or apoptosis (Figure 1). Therefore, the optimization of cold plasma treatment conditions is expected to promote improved skin-wound healing outcomes and inhibit the excessive proliferation of scar tissue.

## 4. Effect of Cold Plasma Treatment on Stem Cell Proliferation

Stem cells have the potential to promote the repair and regeneration of damaged tissue. The accurate regulation of stem cell proliferation, differentiation, and apoptosis is very important for the successful production of regenerative medicine and the performance of tissue engineering. The ability of stem cells to participate in the immune regulation and secretion of cytokines and growth factors highlights them as a potential tool for the successful treatment of chronic wound healing [63]. Previous studies have shown that cold plasma can promote stem cell proliferation and maintain its stemness by changing the stem cell niche interface and directly stimulating the stem cells [38].

### 4.1. Cold Plasma Promotes Stem Cell Proliferation by Changing Niche Interface

The surface properties of biomaterials are very important for cell attachment and proliferation activities [64]. The adhesion of pluripotent stem cells to hydrophobic Petri dishes is poor. Polystyrene, as a hydrophobic material, cannot provide a favorable growth environment for stem cells. However, the surface of polystyrene materials treated with cold plasma becomes hydrophilic, which can provide better conditions for the adhesion, growth, and proliferation of stem cells [65]. Prasertsung et al. [66] used cold plasma (DBD, discharge power varied from 3 to 12 W, a period of time for 9 to 15 s) to treat a gelatin membrane to improve its hydrophilicity and surface energy, which was more conducive to the attachment of bone marrow mesenchymal stem cells (MSTs) and increased cell proliferation. Cámara-Torres et al. [5] observed that 3D-polymer-melt-reinforced scaffolds treated with cold plasma increased the hydrophilicity and protein adsorption characteristics of the scaffold’s surface and improved the distribution and proliferation of MSTs. Therefore, cold plasma can change the biological function of a coating material, enhance the adhesion of stem cells to the Petri dish, and indirectly promote the proliferation of stem cells.

### 4.2. Cold Plasma Promotes Stem Cell Proliferation by Direct Stimulation

MSTs have the potential to differentiate into many cell types (such as adipocytes, osteoblasts, chondrocytes, and neurons); however, it is difficult to maintain their stemness in vitro. As the second messenger in cells, NO can activate nitric oxide synthase through the phosphoinositide 3-kinase (PI3K)/AKT signaling pathway, followed by the activation of the ERK pathway and promotion of stem cell proliferation activity [67]. In a study conducted by Park et al. [68], they observed that, without affecting the stemness of MSTs, cold plasma-producing NO could activate the AKT/ERK pathway and phosphorylate NF-κB, and, furthermore, promote the proliferation of MSTs (Figure 2). In addition, cold plasma treatment altered the cell cycle from the G1 phase to the S phase, which significantly increased the proliferation of MSTs and hematopoietic stem cells [63] (Figure 2). In order to further explore the mechanism of cold plasma on the promotion of stem cell proliferation, Park et al. [69] observed that NO produced in cold plasma-treated medium (plasma jet, voltage at 2.45 kV, a single exposure of 50 s, and 10 exposures of 50 s every hour at 9 h after the initial plasma exposure) significantly up-regulated the expression of mRNA and the proteins of cytokines and growth factors in an epigenetic manner, while it down-regulated the expression of mRNA and the proteins of genes related to the apoptosis pathway. Therefore, cold plasma can promote stem cell proliferation and retain its stemness through both direct and indirect pathways, which is conducive to solving the application problems of stem cells in the fields of regenerative medicine and cell transplantation.

## 5. Effect of Cold Plasma on Cancer Cell Apoptosis

Cancer is one of the leading causes of death worldwide. According to the data released by the International Agency for Research on Cancer of the World Health Organization in the year 2020, there are approximately 19.3 million new cancer cases, of which the case fatality rate accounts for 51.60% annually [70]. At present, cancer is mainly treated by surgery, chemotherapy, and radiotherapy; however, these three methods present some limitations. Recent studies have shown that cold plasma is expected to be used in cancer therapy because it can inhibit the proliferation and induce the cell apoptosis of a variety of malignant cancer cells, and does not damage normal cells [9,10,39,40,41,42,43,44] (Table 2).

The same conditions of cold plasma treatment (plasma jet, voltage ranged from 3 to 5 kV, treatment time of 30 s) significantly reduced the number of lung cancer cells, while there was almost no effect evident on the number of normal bronchial epithelial cells [39]. It was also observed in the studies that the inhibitory effect of cold plasma on cancer cells, such as human non-small-cell carcinoma, human cervical cancer, and human liver fibroblastoma cells, was more significant than that on normal human skin fibroblasts [8]. The inhibitory effect on cancer cells cultured using a cold plasma-treated medium was consistent with that for cancer cells which were directly treated with cold plasma, and the cold plasma inhibited the growth of cancer cells that were cultured with a plasma-treated medium in an intensity- and time-dependent manner [39]. The mechanism of cold plasma inhibiting cancer cell proliferation is related to the imbalance of cancer cell proliferation-related genes, which is induced by cold plasma-producing ROS, whether by directly or indirectly treating the cancer cells using a cold plasma-treated culture medium [8,39,71].

The level of ROS present in cancer cells was higher than that in normal cells and cancer cells were more sensitive to oxidative stress than normal cells [72]. Excessive ROS levels can induce the apoptosis of cancer cells by inducing DNA damage and activating tumor suppressor p53 [72]. Chang et al. [73] observed that the apoptosis of oral squamous carcinomas induced by cold plasma (plasma jet, voltage at 2 kV or 4 kV, treatment time of 1 s) was associated with DNA damage and an increased expression of the ataxia telangiectasia mutation (ATM)/p53 signaling pathway, which caused a G1 phase arrest (Figure 2). Cold plasma and radiotherapy have a synergistic inhibitory effect on cancer cells because ROS produced by cold plasma induces DNA damage and G2/M phase arrest, which, furthermore, leads to the apoptosis of cancer cells [8]. The apoptosis of cancer cells can be activated by the cysteinyl aspartate-specific proteinase (caspase) cascade through the extrinsic tumor necrosis factor (TNF) receptor family pathway or the intrinsic mitochondrial pathway. Studies have shown that ROS produced by cold plasma induces the expression of apoptotic signal-regulated kinase 1 (ASK1) and its downstream p38 MAPK and Jun nuclear kinase (JNK) by activating the TNF pathway, and, furthermore, induces the apoptosis of melanoma cells by initiating caspases 3/7 [74] (Figure 2). Therefore, exogenous excessive ROS produced by cold plasma can induce the apoptosis of cancer cells, and the level of intracellular ROS is also an important factor affecting the processes of cell transduction, proliferation, and death. It has been observed in the research that nicotinamide adenine dinucleotide phosphate oxidase 2 (NOX2) can reduce intracellular oxygen molecules to superoxide anions through the NADPH-dependent single-electron reduction process; thus, NOX2 is the main source of intracellular ROS [75]. Intracellular ROS produced by NOX2 can induce the expression of p38 MAPK by activating ASK [74]. In their study, Ishaq et al. [76] observed that cold plasma treatment (plasma jet, voltage varied from 1.1 to 1.8 kV, treatment time of 30 s) could induce the apoptosis of colorectal cancer cells through the NOX2–ASK1 pathway (Figure 2).

ROS produced by cold plasma can be directly transported into cancer cells. Cold plasma treatment performed beyond certain intensity and time thresholds leads to the inhibition of cancer cell growth and apoptosis through inducing oxidative stress, DNA damage, and apoptosis-related pathways, while causing little or no toxicity to normal cells in the body. Therefore, the optimal doses of cold plasma treatment should be explored even further for its application in the clinical treatment of cancers in the future (Figure 1).

## 6. Potential Application of Cold Plasma in the Field of Animal Husbandry

Animal husbandry is an important part of modern agriculture, which focuses on the animal-breeding environment, animal health, their growth and reproductive performance, and animal food processing and preservation. Several factors, such as environmental pollution and the abuse of antibiotics, have severely affected the progress of animal husbandry, and the health and safety of both animals and human beings. Exploring new technologies and methods to improve animal husbandry practices and ensure the safety of animal food consumption is an urgent requirement for the healthy development of animal husbandry. As a safe and environmentally friendly form of technology, cold plasma has a potential application value in the field of animal husbandry [19].

### 6.1. Cold Plasma Improves Animal Breeding Environments and Health

The presence of a variety of microorganisms and animal excreta in the animal-breeding environment can increase the risk of disease in animals. The purification of their breeding environment can not only provide animals with a healthy and comfortable space, but can also help to reduce the risk of biological pollution. Cold plasma can not only effectively inactivate methicillin-resistant *Staphylococcus aureus*, *Dermatobacter acne*, *Escherichia coli*, and *Salmonella typhimurium* in the environment [18,32], but can also reduce harmful volatile organic compounds and odors in the air [19]. This fact shows that cold plasma has the potential to purify the breeding environments of animals. In addition to indirectly protecting the animals’ health by improving their breeding environments, the disease resistance of animals can be improved by directly injecting the vaccines, where inactivated vaccines play an increasingly important role in the healthy development of animal husbandry practices. At present, cold plasma has been used in the preparation of inactivated vaccines against Newcastle disease and H9N2 influenza viruses. Cold plasma treatment administered for an appropriate amount of time (plasma jet, 2 min) can inactivate the virus without destroying its antigenic determinant. Compared with the traditional inactivated vaccine, which is prepared using formaldehyde, cold plasma can induce higher specific antibody titers for protecting chickens from Newcastle disease and avian influenza [20]. Therefore, cold plasma is expected to be used in the future to improve animal-breeding environments and protect the life and health of animals.

### 6.2. Cold Plasma Improves Animal Growth and Reproduction Outcomes

The growth and reproductive performance of animals are factors directly related to production level and economic development outcomes. In recent years, our laboratory studied the effects and mechanisms of cold plasma on chicken growth and reproduction rates [22,23,24,25,26,27]. We observed that cold plasma treatment administered to chicken HH20-stage embryos at an appropriate intensity (voltage at 11.7 kV) and for an appropriate amount of time (1 min) promoted embryonic development, while the administration of high doses of cold plasma (voltage at more than 11.7 kV, treatment time of 4 min) resulted in the death of chicken embryos through the destruction of the antioxidant defense pathway in a dose-dependent manner, which resulted in an excessive accumulation of ROS and reductions in the numbers of genes and proteins related to embryonic development and the concentration of adenosine triphosphate (ATP) present in skeletal muscles [22]. In addition, we observed that an appropriate amount of cold plasma (voltage at 11.7 kV and treatment time of 2 min) improved the growth of chickens, which showed significant increases in their average daily weight gain and tibia length. The improving mechanism of cold plasma treatment on chicken growth outcomes was related to the increased ATP level and ROS homeostasis, which were mediated by mitochondrial respiratory metabolism and the antioxidant defense system; moreover, cold plasma increased the expressions of thyroid and growth hormones and insulin-like growth factor through regulating the demethylation level of growth-related hormone biosynthesis and energy-metabolism-related genes in the skeletal muscles and thyroids of chickens [23,26].

Semen quality is an important indicator of animal reproductive performance, which is closely related to the fertility, litter size, and survival rate of the offspring [77]. Sperm DNA hypomethylation can promote embryo development and differentiation, while hypermethylation leads to spermatogenesis disorders and sperm quality defects [78]. In vivo and in vitro experiments showed that the appropriate administration of cold plasma treatment (voltages at 11.7 kV, treatment time of 20 s for in vitro spermatozoa and 2 min for in vivo semen) improved male-chicken sperm quality and fertility via the increases in serum testosterone and sperm ATP levels, which were regulated by DNA demethylation, miRNA differential expression, semen ROS homeostasis, and sperm mitochondrial metabolism [24,25,26]. In addition, cold plasma affected the proliferation of chicken Sertoli cells through the adenosine-monophosphate-activated protein kinase-rapamycin target protein signaling pathway, which were target-regulated by miR-7450 and miR-100 in a dose-dependent manner (voltage at 11.7 kV, treatment time of 30 s increased Sertoli cell viability and growth whereas more than 60 s led to cell viability inhibition and cell apoptosis); this was beneficial to provide nutritional and structural support for the spermatogenesis process [27]. Therefore, cold plasma can be administered to improve the growth and reproductive performance of poultry; however, it remains unclear in the literature whether it can successfully promote the growth and reproduction of livestock. The optimization of cold plasma treatment conditions and its mechanisms need to be explored in future research.

### 6.3. Cold Plasma Is Beneficial to the Processing and Preservation of Animal Foods

Fresh animal food easily deteriorates or even rots during storage due to microbial contamination. Therefore, processing fresh animal food by killing microorganisms can extend its shelf life [79]. ROS, RNS, and NO produced by cold plasma play a key role in the inactivation of microorganisms and the formation of nitrite in meat products [21]. As a food additive, nitrite is often used to pickle meat products and prevent the contamination of pathogenic microorganisms, such as *Clostridium botulinum*, *Staphylococcus aureus*, and *Clostridium perfringens* [80]. Cold plasma technology is used in the research, at present, to pickle pork sausages because cold plasma-producing ROS may directly result in microbial death by destroying microbial DNA; on the other hand, nitrite formed by cold plasma treatment can maintain the color, flavor, and lipid oxidation of pork [21]. Cold plasma as a safe and effective form of technology has been applied in research to extend the shelf life of animal food [81]. Studies have shown that cold plasma can effectively inactivate *Listeria monocytogenes*, *Escherichia coli*, and *Salmonella* present in fresh chicken, pork, and beef products [82,83,84]; reduce the *Escherichia coli* in milk [85] and *Salmonella* on the surface of eggs [86]; and does not damage the quality of meat, milk, and eggs [84,85]. In addition, cold plasma technology can be applied for the successful preservation of animal food because of its advantage of sterilization. Studies have shown that the shelf life of chicken breasts packed in bags was effectively extended [81] because of the application of cold plasma (DBD, voltage at 80 kV and treatment time of more than 3 min), which can deactivate the *Escherichia coli* in the bags [87]. Therefore, cold plasma can be applied in the processing of animal foods, extending their shelf lives.

## 7. Conclusions

Cold plasma technology has been increasingly used in biomedical fields and has shown great application potential in the field of animal husbandry. Cold plasma can promote the proliferation of skin-related and stem cells, which is beneficial to wound healing and tissue regeneration processes; moreover, cold plasma can lead to the death of cancer cells, which is conducive to cancer therapy. The effect of cold plasma on cell activity (proliferation or apoptosis) mainly depends on the levels of bioactive substances, such as ROS, RNS, and NO, which are produced in an intensity- and time-dependent manner. Therefore, it is particularly important to control the intensity level and treatment time of cold plasma therapy. Studying the molecular mechanisms of cold plasma regarding the regulation of cell function can provide researchers with a basis for its successful clinical application. In addition, the various applications of cold plasma to the practices of sterilization, biological purification, vaccine preparation, food processing, and improvements to animal growth and reproductive capacity are conducive to promoting the healthy development of animal husbandry practices. However, cold plasma treatment conditions require further optimization and its mechanisms need to be further studied in the research.

## Figures and Tables

**Figure 1 ijms-24-07160-f001:**
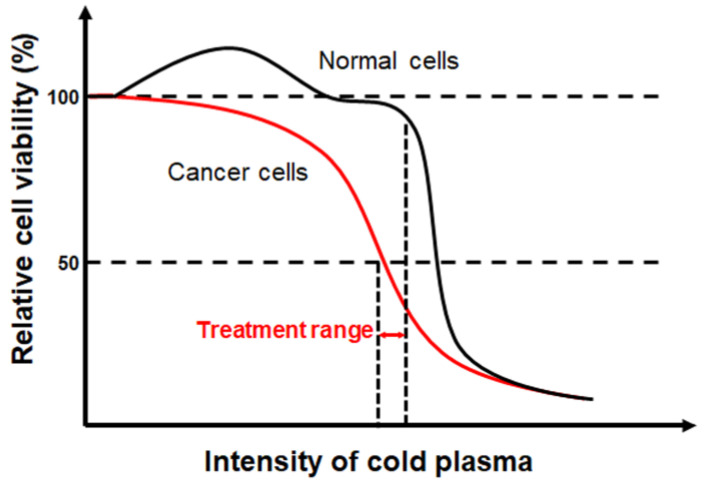
Effects of cold plasma treatment on cell activity [9,10,15,16,39,40,41,42,43,44,53,54]. The appropriate intensity of cold plasma treatment for cancer and normal cells is the condition that inhibits the viability of cancer cells but does not damage the viability of normal cells. Low-intensity cold plasma treatment shows cell viability improvements for normal cells but does not inhibit cancer cell viability. High-intensity cold plasma treatment inhibits cell viability of both normal and cancer cells. For the treatment ranges of cold plasma for different cancer cells see Table 2.

**Figure 2 ijms-24-07160-f002:**
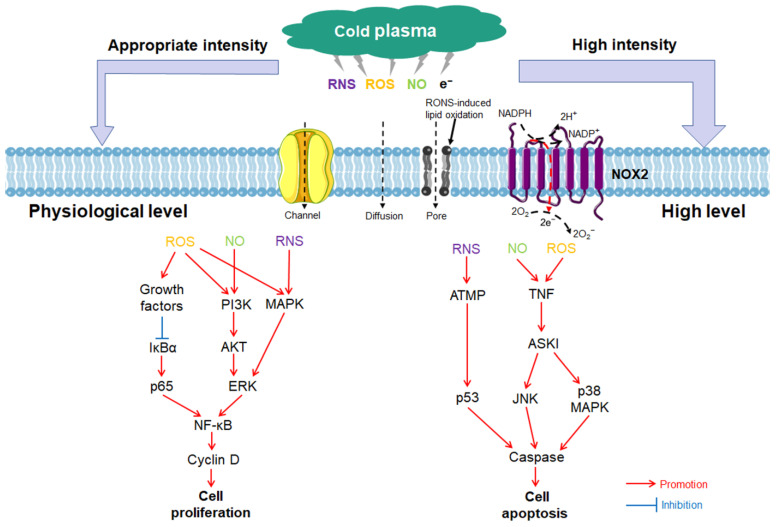
Mechanisms of cold plasma treatment in cell activity. Cold plasma-producing bioactive substances, such as reactive oxygen species (ROS), reactive nitrogen species (RNS), and nitric oxide (NO), affect cell activity in an intensity-dependent manner. Cold plasma-producing reactive oxygen and nitrogen species (RONS), NO, and e^−^ may cross a cell membrane barrier to reach the cell interior through a membrane channel, a passive diffusion process, or pores generated by lipid oxidation. nicotinamide adenine dinucleotide phosphate oxidase 2 (NOX2) can reduce oxygen molecules to superoxide anions through the NADPH-dependent single-electron reduction process. Appropriate intensity cold plasma produces a physiological level of bioactive substances, which may promote cell proliferation through regulating cell cycle progression that is mediated by mitogen-activated protein kinase (MAPK)/extracellular regulated protein kinase (ERK), phosphoinositide 3-kinase (PI3K)/AKT/ERK, and nuclear factor kappa-B (NF-κB) signaling pathways. High-intensity cold plasma generates a high dose of bioactive substances, which may induce cell apoptosis through regulating p53, Jun nuclear kinase (JNK), and p38 MAPK-mediated caspase pathways.

**Table 1 ijms-24-07160-t001:** Applications of cold plasma treatment in regulating cell activity.

Equipment Type	Cell Type	Effect	Application
Plasma jet	Human keratinocytes	Promote proliferation	Repair of skin damage [16]
	Mouse fibroblasts	Promote proliferation	Wound healing [37] and biofunctionalized materials [34,35,36]
	Stem cells	Promote proliferation	Wound healing, tissue engineering, and regenerative medicine [38]
	Human lung cancer cells, human cervical cancer cells, human liver fibroblastoma cells, human brain glioblastoma and colon cancer cells, human bladder cancer cells, human esophageal cancer cells, mouse malignant murine melanoma cells	Inhibit proliferation and promote apoptosis	Cancer therapy [8,9,10,39,40,41]
Dielectric barrier Discharge (DBD)	Porcine aortic endothelial cells	Promote proliferation	Angiogenesis and vascularization of tissue engineering [15]
	Human fibroblasts	Inhibit proliferation	Reduces scar formation [17]
	Human lung cancer cells, human hepatocellular carcinoma cells, human breast cancer cells	Inhibit proliferation and promote apoptosis	Cancer therapy [42,43,44]

**Table 2 ijms-24-07160-t002:** Cold plasma treatment ranges for different cancer cells.

Type of Cancer Cells	Type of Cold Plasma	Treatment Range
Human hepatocellular carcinoma cell lines (Hep3B and Huh7)	DBD	Voltage: 4.7 kV; discharge power: 0.87 W; time: 0.8 μs; distance: 15 mm [42]
Human lung cancer cell line (H460)		Voltage: 80 V; discharge power: 5.7 W; time: 1 to 3 min; distance: 4 mm [43]
Human breast cancer cell line (Hs578T)		Voltage: 27.6 kV; discharge power: 1.8 to 4.8 W; time: 2 min; air flow rate: 1 L/min; distance: 1 mm [44]
Human brain glioblastoma cell lines (U87 MG and U251) and human colon cancer cell line (HCT-116)	Plasma jet	Voltage: 1 kV; current: 100 mA; time: 10 to 180 s; nitrogen flow rate: 1 L/min; distance: 5 mm [9]
Human lung cancer cell line (SW900)		Voltage: 3 to 5 kV; time: 30 s; helium flow rate: 10 to 20 L/min; distance: 1 cm [39]
Mouse malignant murine melanoma cell line (B16F10)		Current: 10 mA; time: 45 s; argon flow rate: 2.5 L/min; distance: 7 mm [10]
Human bladder cancer cell lines (HT-1376 and TCCSUP)		Voltage: 4000 ± 121 V; current: 33 ± 1 µA; time: 15 to 120 s; distance: 2 mm [40]
Human esophageal cancer cell line (KYSE-30)		Voltage: 10 kV; time: 60 s; helium and oxygen mixture flow rate: 2 L/min; distance: 1 cm [41]

## Data Availability

Not applicable.

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
