# Peer review of "The Regulatory Mechanism of Cold Plasma in Relation to Cell Activity and Its Application in Biomedical and Animal Husbandry Practices"

_ijms, 2023, doi:10.3390/ijms24087160_

Round 1

Reviewer 1 Report

I have reviewed the paper "Regulation of cold plasma on cell activity and its application in animal husbandry" and found the paper can be accepted after major revision.

-More physical explanation of results is required.

-The Abstract should be improved.

-The quality of the figures should be improved.

-Finally, the language of the paper needs to be polished.

-Based on the topic the title is so short and needs to clarify the problem statement clearly.

-The analysis quality is too poor please improve the quality and put some arrays on the important part

-The language of the paper needs major polish.

Author Response

-More physical explanation of results is required.

Response: Thank you for your suggestion. We have provided the detailed treatment conditions of cold plasma on different cells and added some physical explanation of relevant results in the revised manuscript. (Changes appear on pages 2-4, 6-11)

-The Abstract should be improved.

Response: Considering the reviewer’s suggestion, we have re-written the Abstract section. (Changes appear on page 1)

-The quality of the figures should be improved.

Response: Based on Figure 1, we have provided Table 2 for summarizing treatment ranges of cold plasma technology in treating different cancer cells (treatment range where cold plasma can inhibit cancer cell viability while is safe for healthy normal cells). In addition, we have modified Figure 2 to make it better readability and more aesthetically pleasing. (Changes appear on Table 2 and Figure 2)

-Finally, the language of the paper needs to be polished.

Response: According to the reviewer’s suggestion, this manuscript has been checked for correct English usage by the professional language editing services of MDPI (English editing invoice: english-63297), and we have polished the language throughout the revised manuscript carefully.

-Based on the topic the title is so short and needs to clarify the problem statement clearly.

Response: We have rewritten the title based on the content and topic of this manuscript. (Changes appear on title)

-The analysis quality is too poor please improve the quality and put some arrays on the important part.

Response: According to the reviewer’s comments, we have provided the detailed information of cold plasma treatment for theoretical support, and added some analysis on the regulatory mechanism of cold plasma on cell activity and cold plasma application parts. (Changes appear on pages 2, 3, 7)

-The language of the paper needs major polish.

Response: According to the reviewer’s suggestion, this manuscript has been checked for correct English usage by the professional language editing services of MDPI (English editing invoice: english-63297), and we have polished the language throughout the revised manuscript carefully.

Special thanks to you for your good comments and suggestions.

Reviewer 2 Report

Review of ijms-2266335

This is a nice comprehensive review of plasma technology for applications in cellular biology, cancer studies, animal growth, reproduction etc., in both aspects of cell stimulation (proliferation) and sterilization (apoptosis). The promising use of plasma technology in the sterilization of food and biological products is finely covered. Furthermore, the performance of plasma in pushing the growth of cells (endothelial, keratinocytes, fibroblasts) is also discussed in good manner. There are several issues to be addressed in order to improve the manuscript even better, as follows:

  1. Plasma polymerization is also utilized to create biofunctionalized surface to promote the growth of fibroblast cells, which is not covered yet in this review. Please kindly refer to:
  • Plasma Processes and Polymers 17 (2020) 1900209 https://doi.org/10.1002/ppap.201900209
  • Biochemical Engineering Journal 78 (2013) 198-204 https://doi.org/10.1016/j.bej.2013.02.022
  • Journal of Polymer Science Part B: Polymer Physics 51 (2013) 1361-1367 https://doi.org/10.1002/polb.23341

  1. Figure 1: Please kindly locate the figure and the figure caption on the same page, and not separated.
  2. Figure 1: Related to this figure, please kindly add a table of performance benchmarks of the plasma technology in treating cancer cells, along with the treatment range (intensity where plasma is safe enough for healthy cells, while able to kill cancer cells).  So, the table must consists of (1) Type of cancer cells, (2) Type of plasma (gas, concentration, and (3) Treatment range (or the process parameters such as flowrate, power, time, pressure, etc.)

  1. References: Please write the page numbers in uniform unabbreviated format. There are some with abbreviated last page, some with abbreviated last page but with the last two digits are kept, and some with unabbreviated page number.  Please therefore revise references number 11, 12, 16, 35, 36, 38, 45, 48, 49, 54, 57, 58, 60, 64, 66, 71.
  2. Please write scientific names in italic. Please revise Reference 66 --> …Campylobacter jejuniSalmonella enterica

Author Response

1.Plasma polymerization is also utilized to create biofunctionalized surface to promote the growth of fibroblast cells, which is not covered yet in this review. Please kindly refer to:

Plasma Processes and Polymers 17 (2020) 1900209 https://doi.org/10.1002/ppap.201900209

Biochemical Engineering Journal 78 (2013) 198-204 https://doi.org/10.1016/j.bej.2013.02.022

Journal of Polymer Science Part B: Polymer Physics 51 (2013) 1361-1367 https://doi.org/10.1002/polb.23341

Response: Thank you for your suggestion. It is really true as reviewer suggested that plasma polymerization can be used to create biofunctionalized surface for promoting fibroblast cell growth, we have added relevant statements in the revised manuscript. (Changes appear on pages 3, 7; Table 1; References [34-36])

2.Figure 1: Please kindly locate the figure and the figure caption on the same page, and not separated.

Response: We have relocated the figure and the figure caption on the same page in the revised manuscript. (Changes appear on page 5)

3.Figure 1: Related to this figure, please kindly add a table of performance benchmarks of the plasma technology in treating cancer cells, along with the treatment range (intensity where plasma is safe enough for healthy cells, while able to kill cancer cells). So, the table must consists of (1) Type of cancer cells, (2) Type of plasma (gas, concentration, and (3) Treatment range (or the process parameters such as flowrate, power, time, pressure, etc.)

Response: According to the reviewer’s suggestion, we have provided Table 2 including type of cancer cells, type of cold plasma, and treatment range, to summarize performance benchmarks and treatment ranges of cold plasma technology in treating different cancer cells, we have presented treatment conditions under which cold plasma only inhibited cancer cell activity while does not affect cell activity of healthy normal cells. (Changes appear on page 8 and Table 2)

4.References: Please write the page numbers in uniform unabbreviated format. There are some with abbreviated last page, some with abbreviated last page but with the last two digits are kept, and some with unabbreviated page number. Please therefore revise references number 11, 12, 16, 35, 36, 38, 45, 48, 49, 54, 57, 58, 60, 64, 66, 71.

Response: We are very sorry for our negligence of the page number format of References. We have made correction throughout References section in the revised manuscript carefully. (Changes appear on References section)

5.Please write scientific names in italic. Please revise Reference 66 --> …Campylobacter jejuni… Salmonella enterica…

Response: According to the reviewer’s comment, we have made correction on this format error in the revised manuscript. (Changes appear on References [79])

Special thanks to you for your good comments and suggestions.

Reviewer 3 Report

This is a very interesting review on the application of cold plasma in treatment of cancer in humans, skin issues, animal husbandry, food processing and preservation etc. This seems like a relatively new field that hasn't been explored much so far. I particularly find interesting the part in the manuscript where the authors discuss about the use of cold plasma in cancer as it doesn't affect other healthy cells unlike radiation therapy that often damages healthy cells along with destroying cancerous cells. There is also very interesting information about how cold plasma technology can increase fertility in animals and also help in better storage of meat, eggs and milk. I am attaching the manuscript here along with my comments. The language seems to be fine and does not require lot of editing. I think overall it is a very interesting review as it has significant information about a field lesser known to us so far. Hence, I endorse acceptance of this manuscript after minor revisions mentioned in the attached file.

Author Response

Response: Thank you for your comments. We have checked the attached file carefully and made corrections according to your suggestions. In addition, we have polished the language throughout the revised manuscript.

Round 2

Reviewer 1 Report

I have reviewed the paper "The regulatory mechanism of cold plasma in relation to cell activity and its application in biomedical and animal husbandry practices" and found the paper can be accepted after major revision.

-The Figures quality are too weak please improve the quality and put some arrays on the important part

The following reference are introduce to compare for the preparation of endotelial.

10.22036/NCR.2020.02.002

https://doi.org/10.1007/s11242-021-01618-x

https://doi.org/10.1007/s12221-021-0347-9

Author Response

-The Figures quality are too weak please improve the quality and put some arrays on the important part

Response: Thank you for your comments. It is really true as reviewer suggested that figure quality need to be improved. We have modified Figures 1 and 2 according to the article content carefully, put some detailed information and arrays in the figure legends and main text, and tried our best to improve the quality of Figures. We hope the revised figures and manuscript will be satisfactory for reviewer. (Changes appear on Figures 1 and 2, Figure legends, and page 2)

The following reference are introduce to compare for the preparation of endothelial.

10.22036/NCR.2020.02.002

https://doi.org/10.1007/s11242-021-01618-x

https://doi.org/10.1007/s12221-021-0347-9

Response: According to the reviewer’s suggestion, we have added some statements by citing these references. (Changes appear on page 4)

Special thanks to you for your good comments and suggestions.

Reviewer 2 Report

Review of ijms-2266335-v2

Thank you for the effort in revising the manuscript significantly, especially the benchmark table. The manuscript can be accepted now.

Author Response

Thank you for the effort in revising the manuscript significantly, especially the benchmark table. The manuscript can be accepted now.

Response: Thank you very much for your reviewing on our article.

Round 3

Reviewer 1 Report

The paper is well as revised therefore can be accepted in this form